# Wettability and Surface Roughness Analysis of Laser Surface Texturing of AISI 430 Stainless Steel

**DOI:** 10.3390/ma15082955

**Published:** 2022-04-18

**Authors:** Edit Roxana Moldovan, Carlos Concheso Doria, José Luis Ocaña, Liana Sanda Baltes, Elena Manuela Stanciu, Catalin Croitoru, Alexandru Pascu, Ionut Claudiu Roata, Mircea Horia Tierean

**Affiliations:** 1Materials Engineering and Welding Department, Transilvania University of Brasov, 29 Eroilor Blvd., 500036 Brasov, Romania; edit.moldovan@unitbv.ro (E.R.M.); baltes@unitbv.ro (L.S.B.); elena-manuela.stanciu@unitbv.ro (E.M.S.); c.croitoru@unitbv.ro (C.C.); alexandru.pascu@unitbv.ro (A.P.); ionut.roata@unitbv.ro (I.C.R.); 2BSH Electrodomésticos España, S.A., Avda. de la Industria 49, 50016 Zaragoza, Spain; carlos.concheso@bshg.com; 3Departamento de Física Aplicada e Ingeniería de Materiales, Universidad Politecnica de Madrid, C/ José Gutiérrez Abascal 2, 28006 Madrid, Spain; joseluis.ocana@upm.es

**Keywords:** surface roughness, wettability, surface laser texturing, surface patterning, ferritic stainless steel

## Abstract

Due to its wide applicability in industry, devising microstructures on the surface of materials can be easily implemented and automated in technological processes. Laser Surface Texturing (LST) is applied to modify the chemical composition, morphology, and roughness of surfaces (wettability), cleaning (remove contaminants), reducing internal stresses of metals (hardening, tempering), surface energy (polymers, metals), increasing the adhesion (hybrid joining, bioengineering) and decreasing the growth of pathogenic bacteria (bioengineering). This paper is a continuation and extension of our previous studies in laser-assisted texturing of surfaces. Three different patterns (crater array-type C, two ellipses at 90° overlapping with its mirror-type B and 3 concentric octagons-type A) were applied with a nanosecond pulsed laser (active medium Nd: Fiber Diode-pumped) on the surface of a ferritic stainless steel (AISI 430). Micro texturing the surface of a material can modify its wettability behavior. A hydrophobic surface (contact angle greater than 90°) was obtained with different variations depending on the parameters. The analysis performed in this research (surface roughness, wettability) is critical for assessing the surface functionality, characteristics and properties of the stainless steel surface after the LST process. The values of the surface roughness and the contact angle are directly proportional to the number of repetitions and inversely proportional to the speed. Recommendations for the use of different texturing pattern designs are also made.

## 1. Introduction

To achieve a successful combination in the joining of dissimilar types of materials is imperative in creating a solid bond and, because the interactions take place on material surfaces, a tool or procedure for creating the bond is needed, and Laser Surface Texturing (LST) offers the perfect outcome. Enhancing fundamental characters, such as sustainability, tribological and biocompatibility, LST offers precise control regarding the main parameters of the characterization of microstructures (shape, roughness, width, depth, size, recast material, etc.). For modifying the surface of materials to obtain microstructures, in previous literature different approaches such as plasma treatment [1], electrochemical procedures [2], and micro-processes such as grinding [3,4], cutting [5], machining by ultrasonic or laser [6,7,8,9,10,11,12,13] and laser chemical processing (combination of chemical cleaning and laser interference lithography) [14] were used. For this research, the LST is preferred since offers a high-quality precision, a way to be automatized, and because nearly any type of material can be used, and with a low cost.

Laser surface texturing (LST) is a method to engender patterns on the surface of materials to accomplish microstructures. LST can remove the material from the surface (by dissolution, evaporation, expulsion and/or melting) to improve joining properties. In manufacturing and previous literature, LST was applied to modify the chemical composition, morphology, and roughness of a surface (wettability), cleaning (remove contaminants), reducing internal stresses of metals (hardening, tempering, etc.), surface energy (polymers and metals), increasing the strength for adhesion (hybrid joining) and so on [15,16,17,18,19,20,21,22,23]. In the application of LST there is a wide variety of equipment and methods that can be used, but the most used is the nanosecond pulsed laser, followed by the picosecond laser and the femtosecond laser.

A current challenge is bioengineering, which is based on combining new materials or improving existing ones with technologies that allow seeding with different types of cells that can then accelerate their growth and adhesion. These support platforms called skeletons must meet several strict conditions, including the precise size of the pores, their spatial distribution and interconnectivity, and geometry. For these reasons, microstructuring is of great interest. The increase of the surface roughness from 0.5 to almost 3 μm was obtained by modifying it using a femtosecond laser irradiation for biopolymer-ceramic composite materials, while increasing the laser energy and the number of applied laser pulses [24]. Plasma activation has also been used to modify polymeric surfaces (PP and PS) to obtain hydrophobic and even superhydrophobic surfaces against pathogenic bacteria [25]. Wetting behavior modification is also achieved by microstructuring some biopolymers providing solutions for tissue engineering scaffolds and biomedical device implants [26]. Laser micromachining is highlighted with real benefits in bioengineering [27,28,29].

Applications of non-wetting surfaces are multiple and are in various areas such as anti-corrosion [30], cleaning [25], self-cleaning [31,32], boiling [33], anti-icing [16,34,35], oil separation [36], textiles [37], and microfluidics [38].

The goal of this research paper continues the studies done in [39] and is focused on the influence of geometry and LST technological paraments on wettability and surface roughness.

## 2. Materials and Methods

### 2.1. Material

The material selected for microstructuring is ferritic stainless steel AISI 430, with low content of carbon and nitrogen, which helps to improve toughness, ductility, and weldability. The ferritic stainless steel has the nominal yield strength 0.2% offset >260 MPa and has 450–600 MPa tensile strength, with elongation >20% at room temperature (20 °C). The material was provided by Acerinox from Madrid, Spain under the name ACX 500. The chemical composition, mechanical and physical properties are described in Table 1 according to technical data received from the manufacturer. ACX 500 (AISI 430) has a good resistance in a wide variety of corrosion environments, (e.g., hydrogen peroxide from 10% (by weight) at 210 °C, nitric acid 40% (by weight) at boiling, sodium hydroxide 20% (by weight) at 500 °C, etc.), and characteristic corrosion rates less than 0.10 mm/year.

The spectral reflectivity of the material sample was analyzed with a GTF Spectrophotometer (Photonic Technologies Group, Woking, UK) before applying the LST. The reflectance of the material surface shows the ability to reflect the radiant energy, and because the material proposed for LST has a bright annealed surface finish, a spectral analysis to choose the parameters for LST (Figure 1) is proposed. Spectral reflectance indicates how much of the incident laser radiation is reflected and absorbed by the material. Because of the high reflectance, the mean power of 20 W for all patterns was maintained constant in order to achieve good results.

### 2.2. Laser Equipment

The equipment used to achieve the microstructuring of stainless steel is TruMark 5020 (Trumpf Laser und Systemtechnik GmbH, Ditzingen, Germany). It can deliver high pulse frequencies, obtaining high performance at increased processing speeds. The laser is generated by an Nd: Fiber active medium, with average power of 20 W and 1064 nm wavelength. The maximum marking area is 180 × 180 mm at 254 mm focal distance and a pulse frequency from 5 kHz (9 ns pulse duration) to 1000 kHz. Because of the specular reflection (surface finish of the ferritic stainless steel is bright annealed), the laser beam axis was settled at 3° angle to prevent possible damage of the laser optical fiber. Throughout the experiments, the constant parameters were the laser power, focal distance, impulses per point, spot diameter, and power density, and the variable parameters were repetition rate, speed, and the number of repetitions, as presented in Table 2.

The reason behind the choice to use a nanosecond pulsed laser instead of the picosecond laser and the femtosecond laser was imposed by the industrial applications of microstructuring, which require the high flow rate, easy automation, and a consideration of the economic demands of the process. In order to achieve a 99% overlap, it is necessary to keep a frequency-speed relationship [40], the speed being directly proportional to the frequency:(1)S=((1−Os/100)d·f)/(1−(1−Os/100)w·f)
where: *S* = speed [mm/min], *Os* = overlap of the spot, *d* = spot diameter [mm], *f* = frequency [Hz], *w* = pulse width [ms].

### 2.3. Microstructuring

Due to the integrated computer-aided design software with the laser system equipment, a wide variety of geometrical shape disclosure can be applied for microstructuring, allowing different patterns to be designed without difficulty. Until now, the most used pattern designs were dimple/hole/crater array [41,42,43,44,45,46,47,48,49], parallel lines [50,51,52,53,54,55,56,57,58,59,60], meshed or cross-pattern [48,53,55,60,61,62,63]. The outstanding pattern designs were square [49,57,62], triangle [62], coral-rock and star-like [64], single ellipse [39,65,66], lotus leaf-like [52], prominent structures [67], octagons [39], and Siberian-Cocklebur-like [34]. This study addresses a new pattern design, three octagonal donuts (type A, Figure 2), two ellipses at 90° angle (type B, Figure 3) and the dimple/hole/crater array (type C, Figure 4). The pattern type C was chosen as a connection and term of comparison between the existing literature and the present paper, the parameters being chosen experimentally. Type B and A were chosen in order to have a wide applicability in industry, from tribological applications to joining dissimilar materials.

The test sample with dimensions of 80 × 25 × 0.5 mm were cut form a single sheet of AISI 430. The steel samples were cleaned and degreased prior to laser processing. The difference between laser textured specimens consists in the area marked on each type of design. For design type A (Figure 2), the marked area is 20 × 19.5 mm, with a 3 mm distance from the edge on the longitudinal way and 2.75 mm edge distance (both edges) on the transversal direction. The same pattern applies for type design B (Figure 3), with a marked area of 18 × 16 mm (longitudinal 3.63 mm and 4.5 mm transversal) and C (Figure 4), with 19.5 × 17 mm area (3.63 mm longitudinal array and 4 mm edge for the transversal direction). Another method of differentiation due to the chosen geometric shape is the center-to-center spacing.

Another important aspect in the LST process is spot overlapping. It was observed that overlapping higher than 90% will behave and form a continuous dent, similar to a seam weld by spot overlapping (Figure 5c). For pattern C the distance between the laser spot centers is 500 µm. For A and B patterns, the laser spot overlapping is the percentage of laser of laser spot coverage of two successive points.

The heat generated by the laser must diffuse efficiently, while the heat captured in the contact area must lead to a greater depth while the affected area must be reduced. The use of low laser fluency is strict for an improved result. The number of spots applied must guarantee the successful texturing. An important phase was to achieve a good repeatability, due to the same trajectory, speed and frequency used. Repeatability was chosen randomly, starting from one pass, then every five repetitions up to a maximum of twenty repetitions (design type A and B) and fifteen repetitions (design type C). The maximum number of repetitions was chosen due to the increase of the splashes and burrs. Increasing the amount of expulsed material will increase the influenced thermal area and the amount of recast material.

### 2.4. Procedure and Analysis Equipment

A roughness tester type ISR-C100 (Insize Co, Suzhou New District, China) (Table 3) was used for surface roughness high accuracy measurement. Because internationally there are used various parameters of roughness, for this research were considered the most important: R_a_—the arithmetical mean of the absolute values of the profile deviations from the mean line of the roughness profile, R_t_—the total height of the roughness profile and R_z_—mean roughness depth [68,69]. The final results were obtained as three values with an average of five measurements for each sample. 

The testing procedure for wettability was performed by dripping a distilled water solution of 10 µL with a micropipette and measuring the contact angle using an Ossila Contact Angle Goniometer (Ossila Ltd., Sheffield, UK) and measurement software Ossila Contact Angle v4.0.0.0 (Ossila Ltd., Sheffield, UK), with 15 s video length and 15 fps frame rate (taking the measurement for each second concluding in an average contact angle measurement).

## 3. Results and Discussion

A significant precision can be achieved with the short pulse laser because the nanosecond pulse duration is proper for surface processing in removing or ablation. The results obtained were chosen from data collected from the experimental results of microstructures created by local ablation. The results offer groove-type sections with different depths and recast material, expelled on the edge of the hollow, generated by the laser beam angle of attack, repetition, power, speed and frequency of the microstructuring parameters. The surface was cleaned with isopropanol (for degreasing and cleaning) before the microstructuring process was applied to homogenize the surface, and kept at room temperature (23 °C). The sketch pattern was designed with the AutoCAD (2D drawing) program. The program is integrated as a part of the laser equipment which had facilitated the drawing transfer.

### 3.1. Surface Morphology after Laser Surface Texturing

Operations applied with a nanosecond pulsed laser, such as melting and ablating, are the mechanisms which are creating the surface modification. Laser ablation is a direct, non-contact surface texturing with high quality, precision and a high level of repeatability and feasibility of the process. Different ranges of parameters (speed, frequency, number of repetitions) were applied, but were kept correlated. Among the results one may distinguish the evolution of the pulse width (time measured at pulse full width half maximum), pulse energy, and fluence (ratio of laser pulse energy and surface area) (Figure 6). Fluence affects the deepness of the ablated thickness and diameter of the ablated crater, both dimensions increasing with the increase in the fluence, as was revealed in [70,71,72]. The ablation rate is also influenced by fluence, and in [73] the direct proportionality was demonstrated. A pulse energy higher than 400 µJ (frequency from 30 to 50 kHz) occurs at 20 W constant mean power, with 3.8 kW pulsed peak power. For the used frequency range (30–100 kHz), fluence decreases in the 8.49 to 2.55 J/cm^2^ domain. This emphasizes a decreasing course for the ablation rate with the increasing of frequency, meaning less material is removed from the surface of the textured material while the frequency increases.

The spots number per cm^2^ are represented in Figure 7, by speed and hatch axis direction. The speed direction is the direction of direct laser texturing, and the hatch direction is the distance between the center to center of the pattern shape applied. Figure 7 outlines that there are more spots created on the velocity direction than the hatch direction, meaning that speed influences the roughness and the number of spots per cm^2^. The spots number can be a decisive factor in defining the necessary resolution for the desired purpose in micro texturing. The number of spots on the hatch direction is constant, different for each kind of pattern. For the speed direction, the number of spots is decreasing while the speed increases.

### 3.2. Surface Roughness

Since direct laser surface texturing is a fast, clean, eco-friendly, and a pliable proceeding in manufacturing for vast geometry patterns (micro- and nano-scale) with high precision, it becomes a reliable source to modify the surface roughness of materials. An important part may be the recast material formation in surface roughness because it is offering an irregular surface and an increased area [39]. Unfortunately, in the research literature there aren’t enough details regarding the impact or resistance of the recast material. In tribological applications, recast material can be an impediment, but the joining of dissimilar materials requires increasing the contact area. It can also have unintended consequences caused by less defined patterns. Surface roughness directly influences wear resistance (the pressure increases due to the contact that is made on the top of the recast material) and corrosion resistance (when subjected to the action of various corrosive environments). 

Surface roughness of the base material is 2.253 µm, (R_z_) 8.828 µm (R_t_) and 0.681 µm (R_a_), the measurement conditions being the same as for the textured specimens. In Figure 8, Figure 9 and Figure 10 one may observe that total height of the roughness profiles (R_t_) reaches 100 µm for design A (10 repetitions/speed 350 mm/s) (Figure 8b), 80 µm for pattern design B (no. of repetition 10/speed 350 mm/s) (Figure 9b) and 90 µm for design C (no. of repetition 15/speed 300 mm/s) (Figure 10b). The measures for the arithmetical mean of the absolute values (R_a_) on the entire surface attain 6.5 µm for design A (20 repetitions/speed 350 mm/s) (Figure 8c), 2.5 µm for design B (20 repetitions/speed 350 mm/s) (Figure 9c) and 4.5 µm for design C (15 repetitions/speed 350 mm/s) (Figure 10c). The mean roughness depth (R_z_) reaches 45 µm for design A (20 repetitions/speed 350 mm/s) (Figure 8a), 19 µm design B (10 repetitions/speed 350 mm/s) (Figure 9a), and 35 µm for design C (15 repetitions/speed 350 mm/s) (Figure 10a).

The total height of the profile (R_t_) from lowest point to highest peak can point out the difference between the sample that has recast material from the expulsed material and the sample that does not. In the results chart for surface roughness for pattern design type A, one may observe that when the number of repetitions is low, the height of the expulsed material as recast material or burr is low. As the number of repetitions of the laser beam transition increases, the height of the expulsed material increases. The difference between the average height and the total height of the profile is the depth of the crevice, and it can be perceived easily from the charts (Figure 8), which applies to all pattern designs. Surface roughness has an influence on function, durability, strength, and manufacturing cost.

It has been observed by surface morphology analysis that surface roughness generally increases with the number of repetitions and decreases when the speed increases. Because spot density on the speed direction also decreases with speed increasing (Figure 7a), it can be concluded that roughness increases in value with spot no./cm^2^ increasing. The pattern applied as micro texturing has an influence on surface roughness as well. For a speed of 300 mm/s and 30 kHz frequency for a single pass, design type A offers the best value of the measured surface roughness (R_a_ = 3 µm), being double in value compared to design type B (R_a_ = 1.5 µm), and one third higher than design type C (R_a_ = 2 µm).

### 3.3. Surface Wettability

Hydrophilicity and hydrophobicity are opposite manifestations of surface wetting. When the liquid droplet spreads (is wetted a large area of the surface) and the contact angle (CA) is less than 90°, the surface is hydrophilic. When the angle is greater than 90°degrees, the surface is hydrophobic. The purpose of this wettability test is to observe whether the textured surface has low surface energy and unitary structures. 

The droplets of distilled water that were placed on the textured surfaces with a micropipette in order to determine the contact angle have a volume of 10 µL. For better accuracy, the procedure was repeated three times for each sample and the result is the average of the measurements. While measuring the contact angle of the laser textured surface of the ferritic stainless steel, a video length duration was set for 15 s, with a frame for every second. The result is for the average contact angle of the measurement for every 15 frames, repeated three. The wetting conditions are described by the surface roughness of the textured samples and can affect the surface free energy by increasing the surface area of the solid phase. The base material, delivered in BA (bright annealed) conditions, has an average contact angle of 42.81°, measured in the same conditions. 

All the patterns applied as texturing on the surface of ferritic stainless steel by a nanosecond pulsed laser point out an improvement regarding the average contact angle of the base material. Considering the chart from Figure 11, the results for design type A, average CA has a constancy, and most of the results are showing a hydrophobicity. The amount of repeatability of the laser beam passing shows an influence. When the number of repetitions is low, CA is less than 90°, indicating hydrophilicity. The influence of the textured pattern points out an improvement (average of the contact angle is double toward base material). It was determined that the contact angle increases at a higher fluence mainly because the fluence is proportional with the repetition rate. Similar results were reported in [72,74]. 

For design type B (Figure 12), the best outcome regarding hydrophobicity involves at least twenty repetitions. When less than twenty repetitions were used on the textured pattern B, hydrophilicity was obtained. Surprisingly, for design type C (Figure 13), the rule that applies is contrary to the other two design types of patterns textured in terms of the number of repetitions of the passes. As the number of repetitions of the micro textured patterns increases, one may observe a slight decrease of the average contact angle. Another notable influence for the average contact angle result is the frequency. When the frequency is lower (applicable for design type B and C) than 40 kHz, the average of the CA doesn’t exceed 90°.

Figure 12 shows two deviations from increased hydrophobicity with the increasing number of repetitions. There is evidence of a deviation from the rule for pattern design type B (the sample with one repetition and 10, 40 kHz frequency), and the distilled water drop is spreading very fast, having a high hydrophilicity wetting behavior. The very rapid spread (after three seconds) of the drop of distilled water and the high wetting rate from hydrophobic to hydrophilicity are noteworthy. An example of these types of modifications in contact angle can be observed in Figure 14 and Figure 15. The result of contact angle measurements for design type B (no. of repetitions 1/ frequency 40 kHz) can’t be measured by the equipment, meaning the microstructures belong to the extreme zone of superhydrophilicity (θ = 0° ÷ 20°).

The results of the wettability analysis of the micro textured areas offer a view on the influence of the number of repetitions and the frequency. As the number of repetitions increases, keeping the frequency constant, the measured CA increases, offering hydrophobicity to the micro-structure’s surface, and a decrease in values of the CA occurs as the frequency increases and the number of repetitions is constant. For a speed of 300 mm/s, with 30 kHz frequency and five repetitions, and the pattern design type A (CA = 92.76°) and type C (CA = 126.9°), the wettability measurement offers hydrophobic surfaces (CA > 90°). The results for a speed of 300 mm/s, frequency 30 kHz and five repetitions with design type B is at the limit of hydrophobia (CA = 79.25°), but with a remarkably improved CA value compared to the base material without LST (CA = 42.81°).

## 4. Conclusions

Surface texturing in the form of groove-type sections with different depths and recast material, expelled on the edge of the hollow, were obtained on AISI 430 stainless steel in three variants of design: three concentric octagonal donut patterns (type A), two ellipses at 90° angle (type B), and a dimple/hole/crater array (type C). 

The morphology of the textured surfaces (in terms of geometrical pattern and roughness) was linked to the laser processing parameters (spot density, processing speed) and surface wettability (contact angle).

It was observed that the roughness of the textured surface is directly proportional to the spot density (number of repetitions), while it is inversely proportional to the processing speed. Surfaces with high roughness were obtained when the recast material was present in significant amounts (design A) due to a higher degree of overlapping when a high crevice depth was registered in the pattern with no overlapping and recast material, as with design type C. Design B has a lower degree of overlapping, therefore an intermediate value was obtained for the surface roughness. Fine-tuning the roughness of stainless steel finds applicability in the industry. The microtextured design type A can be easily applied in joining dissimilar materials due to an irregular surface and an increased contact area. In tribological applications, recast material can be an impediment, but in the joining of dissimilar materials, this increasing of the contact area is required. The pattern design type C is suitable mainly for tribological applications since the recast material is missing. 

The recast material and the crevice depth of the pattern have a marked influence on the contact angle as well. The increase in roughness due to the surface texturing increases the contact angle with values ranging from 30 to 195% higher than for the neat substrate, thereby increasing hydrophobicity. A similar trend was reported in [75]. 

Further investigations will be conducted to achieve super hydrophobic surfaces using the design type B optimized by diminishing the center-to-center distance of the micro texturing pattern.

## Figures and Tables

**Figure 1 materials-15-02955-f001:**
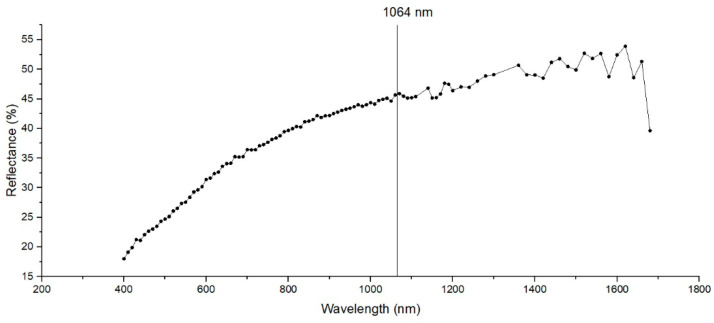
Spectral reflectance of the sample of AISI 430 stainless steel ferritic.

**Figure 2 materials-15-02955-f002:**
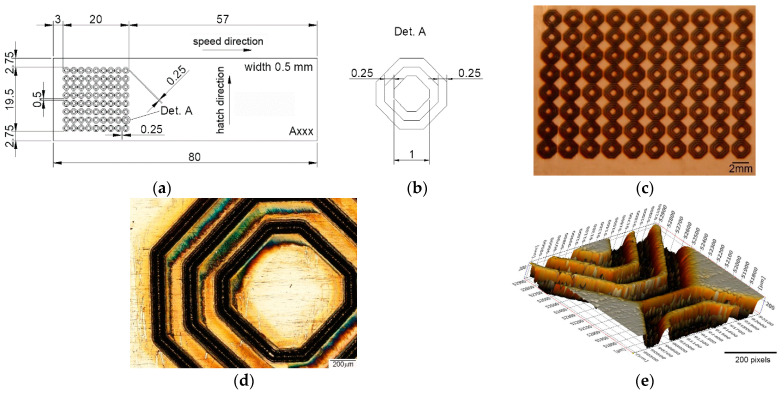
(**a**) Schematic representation of the test sample and pattern design type A, (**b**) detail of the pattern design A (all dimensions in mm), (**c**,**d**) macro and microscopic image, (**e**) 3D topographic image of LST design type A adapted from [39].

**Figure 3 materials-15-02955-f003:**
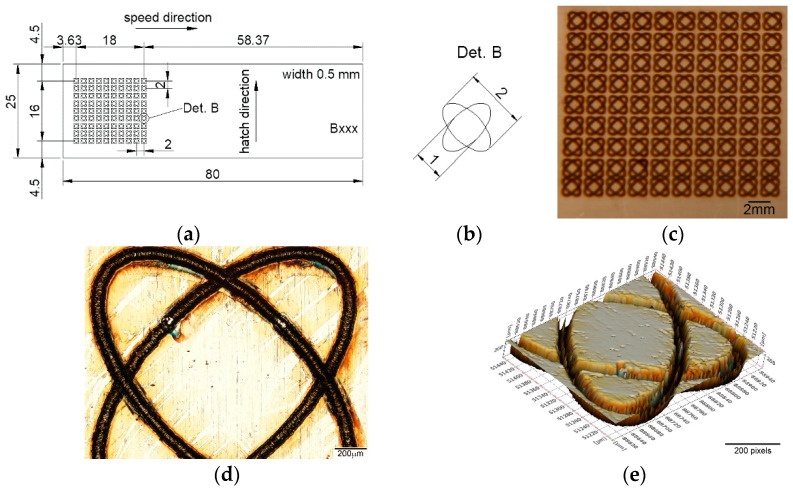
(**a**) Schematic representation of the test sample and pattern design type B, (**b**) detail of the pattern design B (all dimensions in mm), (**c**,**d**) macro and microscopic image, (**e**) 3D topographic image of LST design type B adapted from [39].

**Figure 4 materials-15-02955-f004:**
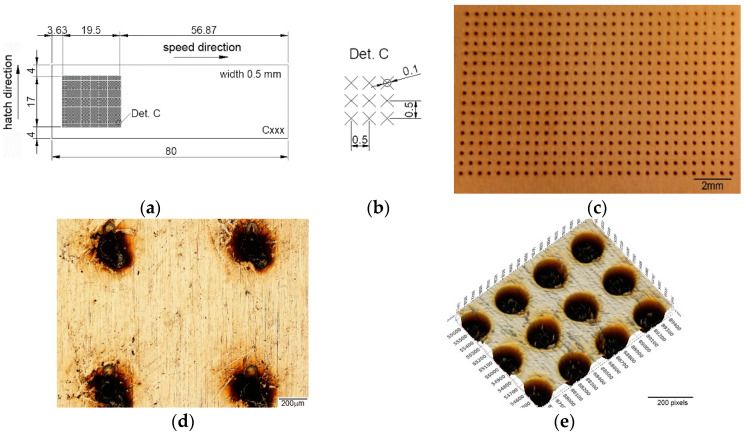
(**a**) Schematic representation of the test sample and pattern design type C, (**b**) detail of the pattern design C (all dimensions in mm), (**c**,**d**) macro and microscopic image, (**e**) 3D topographic image of LST design type C adapted from [39].

**Figure 5 materials-15-02955-f005:**
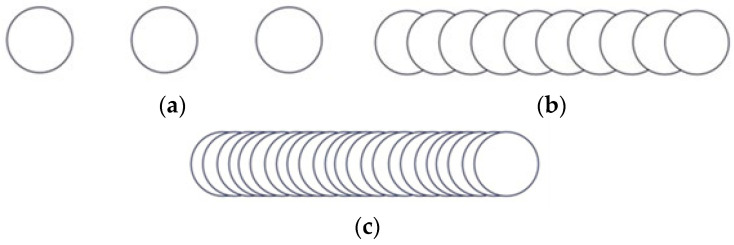
Representation of laser spot overlapping 0% (**a**), 50% (**b**) and 90% (**c**).

**Figure 6 materials-15-02955-f006:**
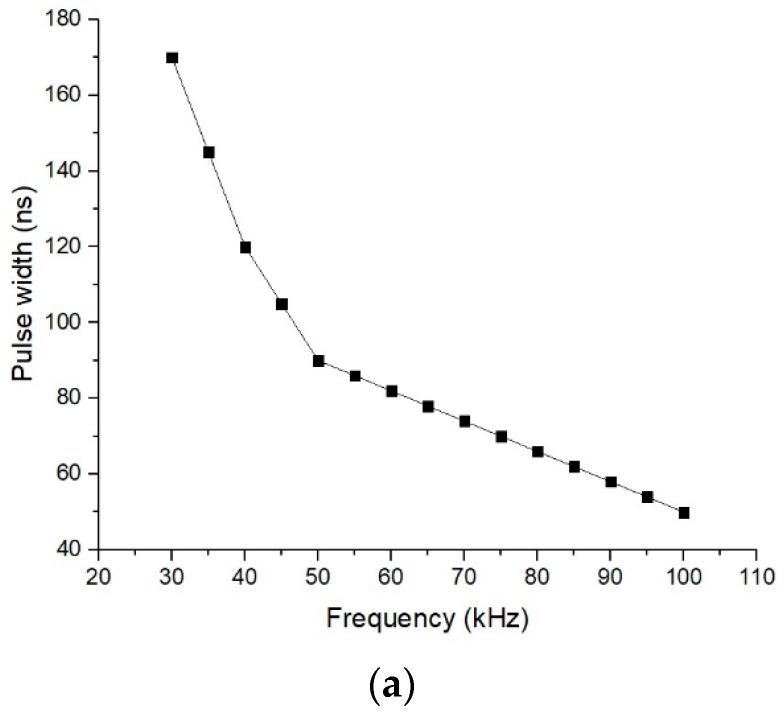
Pulse width (**a**), pulse energy (**b**) fluence (**c**) engendered by frequency.

**Figure 7 materials-15-02955-f007:**
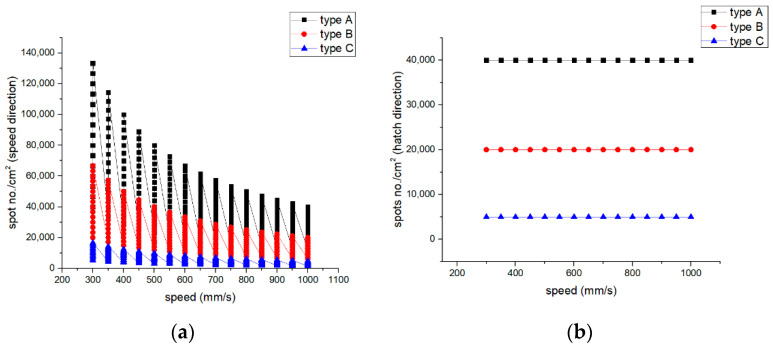
Spots number/cm^2^ vs. speed—(**a**) on speed direction and (**b**) on hatch direction.

**Figure 8 materials-15-02955-f008:**
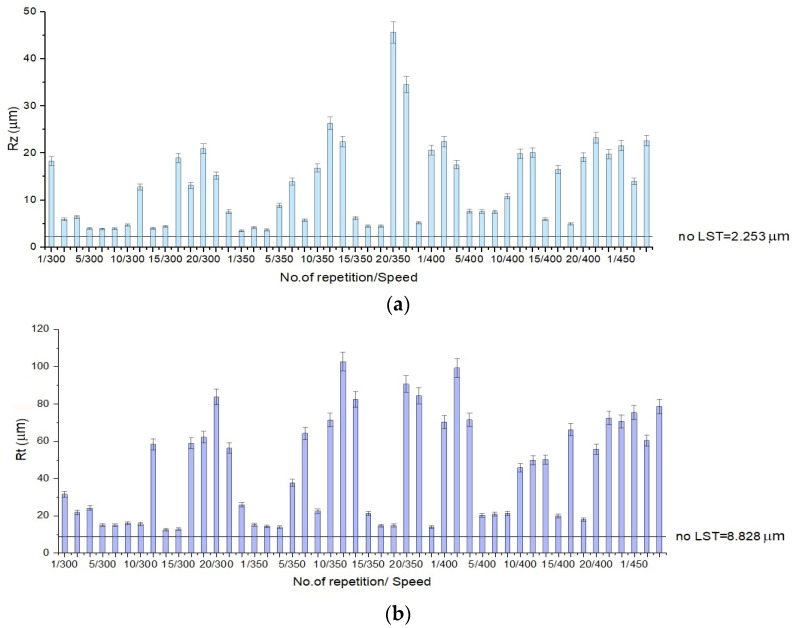
Surface roughness charts for design type A, R_z_—deviation from the mean line, specifically focusing on the highest peak and valley (**a**), R_t_—total height of the roughness profile of the deepest valley within the evaluation length (**b**) and R_a_—measures the average length between the peaks and valleys and the deviation from the mean line on the entire surface (**c**).

**Figure 9 materials-15-02955-f009:**
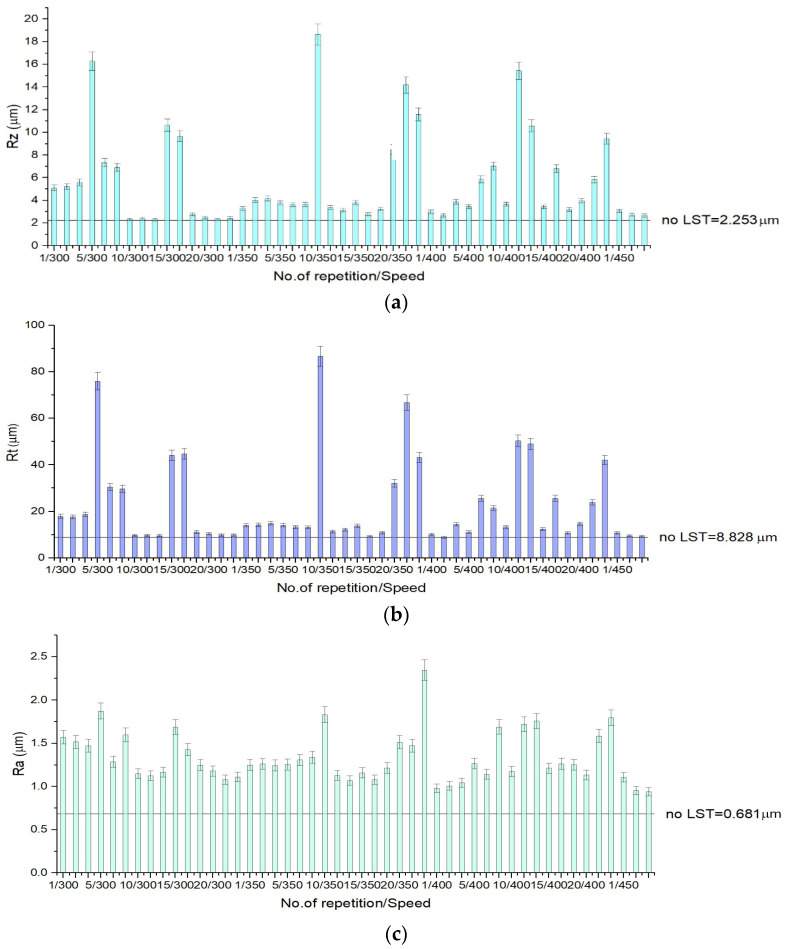
Surface roughness charts for design type B, R_z_—deviation from the mean line, specifically focusing on the highest peak and valley (**a**), R_t_—total height of the roughness profile of the deepest valley within the evaluation length (**b**) and R_a_—measures the average length between the peaks and valleys and the deviation from the mean line on the entire surface (**c**).

**Figure 10 materials-15-02955-f010:**
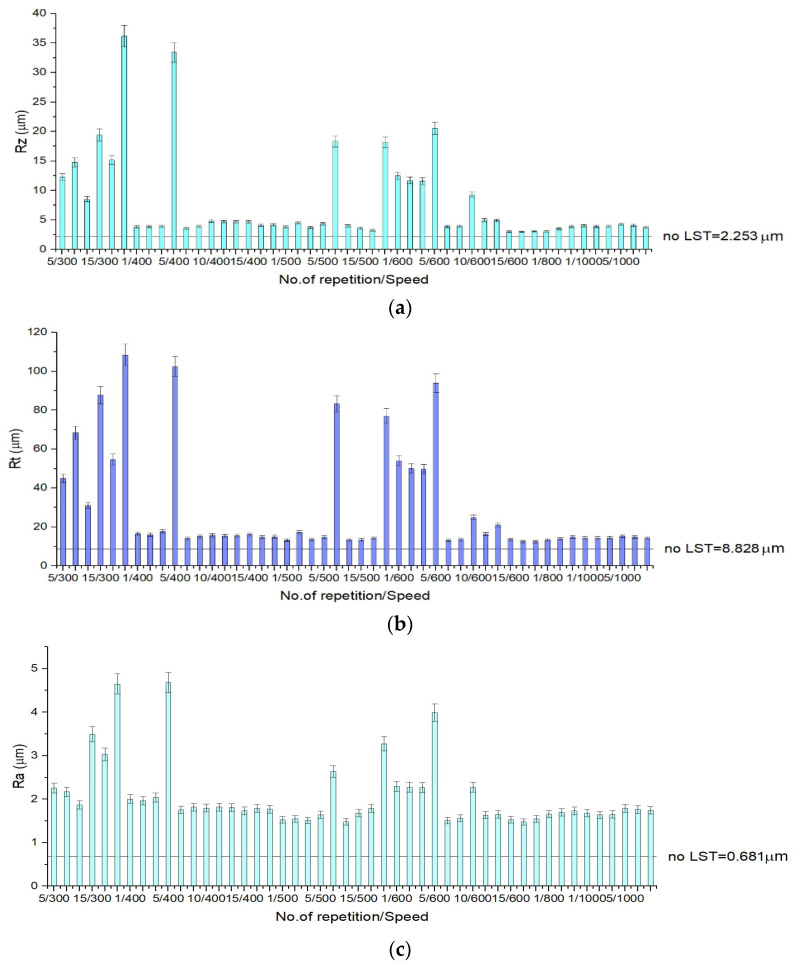
Surface roughness charts for design type C, R_z_—deviation from the mean line, specifically focusing on the highest peak and valley (**a**), R_t_—total height of the roughness profile of the deepest valley within the evaluation length (**b**) and R_a_—measures the average length between the peaks and valleys and the deviation from the mean line on the entire surface (**c**).

**Figure 11 materials-15-02955-f011:**
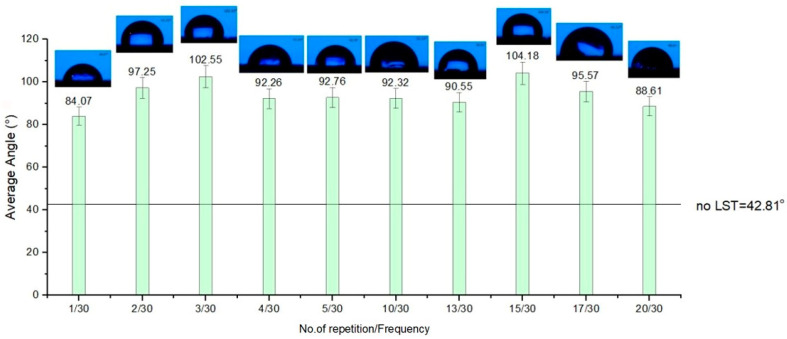
Average contact angle for design type A.

**Figure 12 materials-15-02955-f012:**
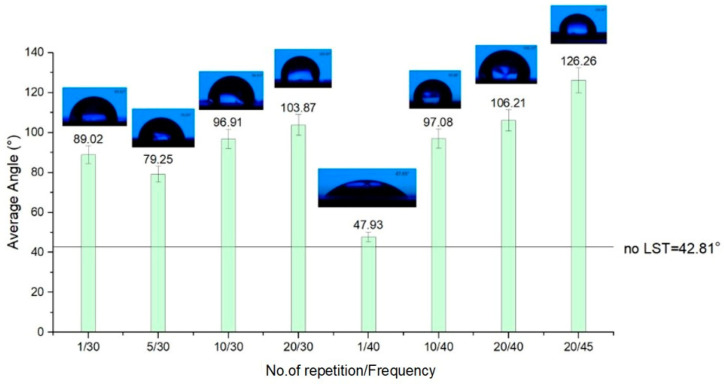
Average contact angle for design type B.

**Figure 13 materials-15-02955-f013:**
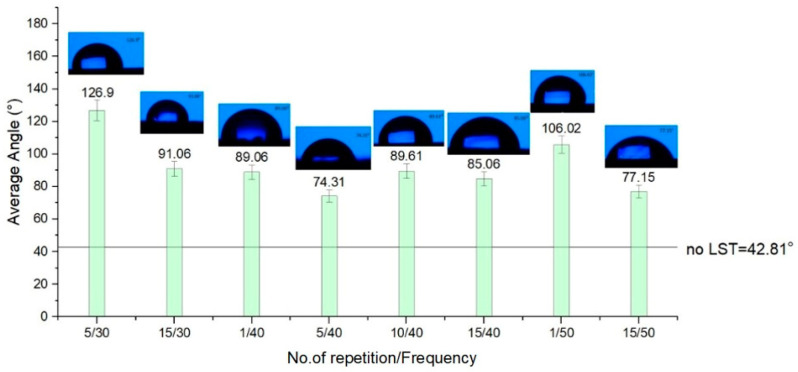
Average contact angle for design type C.

**Figure 14 materials-15-02955-f014:**
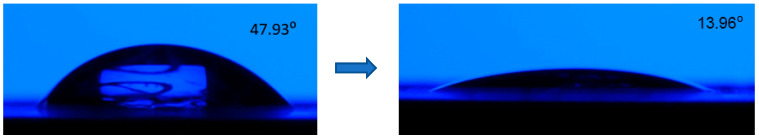
Aberration in contact angle measurement of the sample design type B (no. of repetition 1/frequency 40 kHz).

**Figure 15 materials-15-02955-f015:**
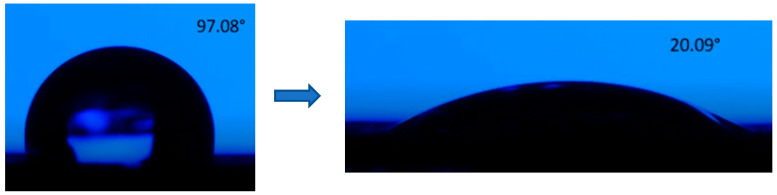
Aberration in contact angle measurement of the sample design type B (no. of repetition 10/frequency 40 kHz).

**Table 1 materials-15-02955-t001:** Chemical composition, mechanical and physical properties of AISI 430 stainless steel, according to ASTM A-240 and EN 10088-2:2005.

Properties	Chemical Element		Concentration (%)
Chemical composition	Carbon (C)		≤0.80
	Silicon (Si)		≤1.00
	Manganese (Mn)		≤1.00
	Phosphorous (P)		≤0.040
	Sulphur (S)		≤0.015
	Chromium (Cr)		16.00–18.00
	Nitrogen (N)		≤0.045
	**Properties**		**Unit Measure**
Mechanical properties	Tensile strength		450–600 MPa
	Proof stress 0.2%		min. 260 MPa
Physical properties	Modulus of elasticity	tension	200 GPa
		torsion	65 GPa
	Density		7.75 g/cm³
	Melting point		1425–1510 °C
	Thermal expansion		10.4 × 10^−6^ /K

**Table 2 materials-15-02955-t002:** Variable and constant parameters for applied LST pattern on AISI 430 stainless steel.

Constant Parameters	Unit Measure	Value
Power	[W]	20
Spot diameter	[µm]	100
Power density	[W/cm^2^]	2.55 × 10^5^
Impulses per point	[number]	1
Track width of the spot	[mm]	0.5
Overlap of the spot	[%]	99
Hatch	[mm]	0.25/0.5/2
**Variable Parameters**		
Frequency	[kHz]	30–100
Speed	[mm/s]	300–1000
Pulse width	[ns]	170–50
No. of repetitions	[number]	1/5/10/15

**Table 3 materials-15-02955-t003:** Technical parameters of equipment ISR-100 used for surface roughness analysis.

Technical Parameters	Unit Measure	Value
Traverse speed	[mm/s]	5
Measuring force	[mN]	4
Accuracy	[%]	±10
Resolution (for R_a_)	[µm]	0.001
Range	[µm]	160

## Data Availability

Data sharing not applicable.

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
