# Peer review of "Wettability and Surface Roughness Analysis of Laser Surface Texturing of AISI 430 Stainless Steel"

_materials, 2022, doi:10.3390/ma15082955_

Round 1

Reviewer 1 Report

I have some questions about this article:

  1. Please briefly explain why nanosecond is used and what advantages it has over femtosecond and picosecond in this experiment, so that the reader can understand more clearly.
  2. The reason for choosing these three experimental texture patterns is that they are obtained by reading other literatures or by testing their own experimental parameters. What are the advantages of rotating these three texture patterns?
  3. The results and discussion in the third part do not clearly show any help to the experimental content, so I am confused as a reader.
  4. It is not clear what role the number of spots per square meter plays in the surface topography after laser surface texturing in 3.1, and what effect the number will have.
  5. In 3.2 surface roughness, the difference between the total contour height and the average contour height can be obtained. What is the impact on the surface structure of the material? Without any speculation or exploration, it is simply summarized as "suitable pattern design".
  6. The final conclusion is rather vague and superficial, and does not clearly explain the influence of several different texture treatment methods on surface roughness and wettability, but simply shows that there is correlation between A and B textures.
  7. The content of the article does not have too many innovative points, the experiment process is relatively simple.

Author Response

We appreciate your constructive comments and have modified the manuscript accordingly. In the revised version the amendments and corrections made to the first submission have been highlighted. All lines indicated below are referred to the Track changes.doc file.

  1. Please briefly explain why nanosecond is used and what advantages it has over femtosecond and picosecond in this experiment, so that the reader can understand more clearly.

In chapter 2.2 were added the lines 122-129:

The reason behind the choice to use a nanosecond pulsed laser instead of the picosecond laser and the femtosecond laser was imposed by the industrial applications of microstructuring, which require the high flow rate, easy automation, and considering the economic demands of the process. In order to achieve a 99% overlap, it is necessary to keep a frequency-speed relationship [40], the speed being directly proportional to the frequency:

S=((1-Os/100)d∙f)/(1-(1-Os/100)w∙f)

(1)

where: S = speed [mm/min], Os = overlap of the spot, d = spot diameter [mm], f = frequency [Hz], w = pulse width [ms].

  1. The reason for choosing these three experimental texture patterns is that they are obtained by reading other literatures or by testing their own experimental parameters. What are the advantages of rotating these three texture patterns?

In chapter 2.3 were added the lines 140-144:

The pattern type C was chosen as a connection and term of comparison between the existing literature and the present paper, the parameters being chosen experimentally. Type B and A were chosen in order to have a wide applicability in industry, from tribological applications to joining dissimilar materials.

The advantages of choosing these three patterns are explained on chapter 4, lines 396-402:

Fine tuning of the superficial roughness and wettability of stainless steels finds applicability in the industry. According to the results, it is pointed out that the microtextured design type A can be easily applied in joining dissimilar materials due to an irregular surface and an increased contact area. In tribological applications, recast material can be an impediment, but in joining of dissimilar materials, this increasing of the contact area is required. The pattern design type C is suitable mainly for tribological applications, since recast material is missing.

  1. The results and discussion in the third part do not clearly show any help to the experimental content, so I am confused as a reader.

The results and discussions were mentioned together in the third part, in order to draw more convincing conclusions for each testing method. Conclusions chapter contains correlation between results obtained in the previous chapter.

  1. It is not clear what role the number of spots per square meter plays in the surface topography after laser surface texturing in 3.1, and what effect the number will have.

The comment from chapter 3.1 has been completed and the new form please find below (lines 230-235):

Figure 7 outlines that there are more spots created on the velocity direction than hatch direction, meaning that speed influences the roughness and the number of spots per cm². The spots number can be a decisive factor in defining the necessary resolution for the desired purpose in micro texturing. The number of spots on the hatch direction is constant, different for each kind of pattern. For the speed direction, the number of spots is decreasing while the speed increases.

  1. In 3.2 surface roughness, the difference between the total contour height and the average contour height can be obtained. What is the impact on the surface structure of the material? Without any speculation or exploration, it is simply summarized as "suitable pattern design".

In chapter 3.2 were added the lines 283-290:

Generally, it has been observed by surface morphology analysis that surface roughness increases with the number of repetitions and decreases when the speed increases. Because spots density on the speed direction also decreases with speed increasing (Figure 7.a), it can be concluded that roughness increases in value with spot no./cm2 increasing. The pattern applied as micro texturing has an influence of surface roughness too. For a speed of 300 mm/s and 30 kHz frequency for a single pass, design type A offers the best value of the measured surface roughness (Ra = 3 µm), being double in value compared to design type B (Ra = 1.5 µm), and one third higher than design type C (Ra = 2 µm).

  1. The final conclusion is rather vague and superficial, and does not clearly explain the influence of several different texture treatment methods on surface roughness and wettability, but simply shows that there is correlation between A and B textures.

Chapter 4 was almost full reformulated. The influence of processing parameters on surface roughness and wettability is explained in the lines 385-402:

The analysis of the wettability and surface roughness reveals a correlation between textured pattern design A and textured pattern design B in terms of increased contact angle and a greater surface roughness due to the recast material deposited at the edge of the crevice. Between pattern design type C and A, the link is created due to high depth of the crevice. The correlation between surface roughness and wettability can be established through frequency, speed, and number of repetitions. The higher the number of repetitions, the higher the values for the surface roughness and the contact angle at constant frequency and speed. But when the frequency and speed increase, and the number of repetitions remains constant, lower values result for surface roughness and contact angle. As consequence, the increase in roughness due to the surface texturing increases the contact angle. A similar trend was reported in [75].

Fine tuning of the superficial roughness and wettability of stainless steels finds applicability in the industry. According to the results, it is pointed out that the microtextured design type A can be easily applied in joining dissimilar materials due to an irregular surface and an increased contact area. In tribological applications, recast material can be an impediment, but in joining of dissimilar materials, this increasing of the contact area is required. The pattern design type C is suitable mainly for tribological applications, since recast material is missing.

  1. The content of the article does not have too many innovative points, the experiment process is relatively simple.

This research is the result of an industrial requirement, and the innovation consists in the design of pattern A and pattern B, as well as the recommendations that are made for possible industrial applications.

Reviewer 2 Report

The authors have done a great job. In my opinion, the material is presented well and understandably, but there are small mistakes that need to be corrected before the publication of this work: 

  1. The Abstract needs to be redone. I offer the following formulation: "This paper is a continuation and extension of our previous studies in laser-assisted texturing of surfaces. Three different patterns (dimple/hole/crater array-type C, ellipses at angle of 45° overlapping with its mirror-type B and 3 concentric octagons-type A) were applied with nanosecond pulsed laser (active medium Nd: Fiber Diode-pumped) on the surface of a ferritic stainless steel (AISI 430). Micro texturing the surface of a material can modify its wettability behavior. A hydrophobic surface (contact angle greater than 90°) was obtained with different variations depending on the parameters. The analysis performed in this research (surface roughness, wettability) is critical for assessing the surface functionality, characteristics and properties of the stainless steel surface after LST process."
  2. Figure 1. Change wavelenght on Wavelenght.
  3. Figure 3. Make the scale marker in Figure (e) clearly readable.
  4. Figure 4. Make the scale marker in Figure (e) clearly readable.
  5. Figure 6 is not readable. You need to improve the quality of this figure. 
  6. Figure 7 (b). Remove the orange square.
  7. The quality of Figures 8-10 needs to be improved.
  8. The conclusions need to be shortened. They need to be presented more clearly. It is better to describe them point by point. I advise to make a separate discussion part.  

Author Response

We appreciate your constructive comments and have modified the manuscript accordingly. In the revised version the amendments and corrections made to the first submission have been highlighted. All lines indicated below are referred to the Track changes.doc file.

  1. The Abstract needs to be redone. I offer the following formulation: "This paper is a continuation and extension of our previous studies in laser-assisted texturing of surfaces. Three different patterns (dimple/hole/crater array-type C, ellipses at angle of 45° overlapping with its mirror-type B and 3 concentric octagons-type A) were applied with nanosecond pulsed laser (active medium Nd: Fiber Diode-pumped) on the surface of a ferritic stainless steel (AISI 430). Micro texturing the surface of a material can modify its wettability behavior. A hydrophobic surface (contact angle greater than 90°) was obtained with different variations depending on the parameters. The analysis performed in this research (surface roughness, wettability) is critical for assessing the surface functionality, characteristics and properties of the stainless steel surface after LST process."

The Abstract has been adjusted as indicated, please find lines 20-28:

This paper is a continuation and extension of our previous studies in laser-assisted texturing of surfaces. Three different patterns (crater array-type C, ellipses at angle of 45° overlapping with its mirror-type B and 3 concentric octagons-type A) were applied with nanosecond pulsed laser (active medium Nd: Fiber Diode-pumped) on the surface of a ferritic stainless steel (AISI 430). Micro texturing the surface of a material can modify its wettability behavior. A hydrophobic surface (contact angle greater than 90°) was obtained with different variations depending on the parameters. The analysis performed in this research (surface roughness, wettability) is critical for assessing the surface functionality, characteristics and properties of the stainless steel surface after the LST process.

  1. Figure 1. Change wavelenght on Wavelenght.
  2. Figure 3. Make the scale marker in Figure (e) clearly readable.
  3. Figure 4. Make the scale marker in Figure (e) clearly readable.
  4. Figure 6 is not readable. You need to improve the quality of this figure. 
  5. Figure 7 (b). Remove the orange square.
  6. The quality of Figures 8-10 needs to be improved.

Figures 1, 2, 3, 4, 6-10 have been remade considering the above observations. In order to increase the readability, some of the figures were arranged in columns.

  1. The conclusions need to be shortened. They need to be presented more clearly. It is better to describe them point by point. I advise to make a separate discussion part.  

The results and discussions were mentioned together in the third part, to draw more convincing conclusions. The final conclusions were reinforced with a few observations, correlating the obtained results.

Reviewer 3 Report

Wettability and Surface Roughness Analyze of Laser Surface Texturing of AISI 430 Stainless Steel (Manuscript ID: materials-1637203). It investigated the effects of three different patterns produced by nanosecond pulsed laser on the hydrophobicity of a ferritic stainless steel. While, some questions should be well illuminated before the manuscript being further processed.

  1. It is stated in the Abstract that this paper investigated the effect surface topology on the surface and wettability. But no specific results or findings were offered in the abstract, which decreases the interest of readers for further reading or realizing the novelty of this paper.
  2. 27 papers published in recent five years were cited in this paper, which is less than half of the total citations. The timeliness of this work should be enhanced through overview more recently published papers.
  3. BTW, the physical influence mechanism of the surface topology on the surface and wettability should be well discussed to highlight the novelty, a related referenced is suggested that can help (International Journal of Hydrogen Energy 46 (2021) 26489-26498).
  4. What is the unit of value in Figure 2(b), 3(b) 4(b), and the scale bar should be offered in Figure 2(c) (d), figure 3(c) (d) and figure 4(c) (d)?
  5. The data displayed in Figure8, 9, and 10 as well as the statement in line 160 that “the average arithmetic of measurements of surfaces peaks and valleys (Ra), the total height (Rt) and the average of the absolute values on height and depth (Rz) will be considered” and line 219 that “Surface roughness of the base material is 2.253 µm (Rz) 8.828 µm (Rt) and 0.681 µm (Ra)”. Why are these three parameters selected? And what is the relationship between them since the value differs significantly.
  6. Following the last question, how do the surface roughness of the base material 2.253 µm (Rz) 8.828 µm (Rt) and 0.681 µm (Ra) obtained? As it is displayed in Fig. 8-10 the raw data of roughness, an average roughness with a standard deviation ishould be offered to make comparisons between these three samples, and the author should offer how is the average roughness obtained from the raw data?
  7. It is stated in line 259 that “For better accuracy the procedure was repeated three times for each sample and the result is the average of the measurements.” My suggestion is that a standard deviation value should be offered with the average WCA value in figure 11, 12, and 13.
  8. Following the last question, why some WCA picture is offered while some is missing?
  9. In the conclusion section, the effect of surface topology on the roughness and wettability should be highlighted since the aim of this paper is to investigate this relationship. Moreover, the data and results should be discussed and concluded sharply and a clear conclusion should be offered.

Author Response

We appreciate your constructive comments and have modified the manuscript accordingly. In the revised version the amendments and corrections made to the first submission have been highlighted. All lines indicated below are referred to the Track changes.doc file.

  1. It is stated in the Abstract that this paper investigated the effect surface topology on the surface and wettability. But no specific results or findings were offered in the abstract, which decreases the interest of readers for further reading or realizing the novelty of this paper.

Partially the Abstract has been modified and reformulated, adding lines 28-30:

The values of the surface roughness and the contact angle are directly proportional to the number of repetitions and inversely proportional to the speed. Recommendations for using of different texturing pattern design are also made.

  1. 27 papers published in recent five years were cited in this paper, which is less than half of the total citations. The timeliness of this work should be enhanced through overview more recently published papers.

The references have been updated and now of the 75 references of the paper, only 10 are older than 2016.

  1. BTW, the physical influence mechanism of the surface topology on the surface and wettability should be well discussed to highlight the novelty, a related referenced is suggested that can help (International Journal of Hydrogen Energy 46 (2021) 26489-26498).

The indicated paper was considered in position [75] of the references. The results obtained in our paper are consistent with the results of Wang et al., please see line 395:

A similar trend was reported in [75].

  1. What is the unit of value in Figure 2(b), 3(b) 4(b), and the scale bar should be offered in Figure 2(c) (d), figure 3(c) (d) and figure 4(c) (d)?

All observations related to the above figures were considered.

  1. The data displayed in Figure8, 9, and 10 as well as the statement in line 160 that “the average arithmetic of measurements of surfaces peaks and valleys (Ra), the total height (Rt) and the average of the absolute values on height and depth (Rz) will be considered” and line 219 that “Surface roughness of the base material is 2.253 µm (Rz) 8.828 µm (Rt) and 0.681 µm (Ra)”. Why are these three parameters selected? And what is the relationship between them since the value differs significantly.

Since the parameters (Ra, Rt, Rz) are used preferentially in different geographical areas, all three were used. The definitions and relationships between these parameters are mentioned in their related standards, references [68,69].

  1. Following the last question, how do the surface roughness of the base material 2.253 µm (Rz) 8.828 µm (Rt) and 0.681 µm (Ra) obtained? As it is displayed in Fig. 8-10 the raw data of roughness, an average roughness with a standard deviation ishould be offered to make comparisons between these three samples, and the author should offer how is the average roughness obtained from the raw data?

In chapter 3.2 was added the remark from line 250:

Surface roughness of the base material is 2.253 µm, (Rz) 8.828 µm (Rt) and 0.681 µm (Ra), the measurement conditions being the same as for the textured specimens.

We modified the figures 8-10 according to the indication received, introducing no LST roughness and error bar.

  1. It is stated in line 259 that “For better accuracy the procedure was repeated three times for each sample and the result is the average of the measurements.” My suggestion is that a standard deviation value should be offered with the average WCA value in figure 11, 12, and 13.

We modified the figures 11-13 according to the indication received, introducing no LST contact angle and error bar.

  1. Following the last question, why some WCA picture is offered while some is missing?

All of the WCA pictures were updated in the figures 11-13.

  1. In the conclusion section, the effect of surface topology on the roughness and wettability should be highlighted since the aim of this paper is to investigate this relationship. Moreover, the data and results should be discussed and concluded sharply and a clear conclusion should be offered.

The conclusion section was almost full reformulated. Please see lines 363-404:

The present study demonstrates that high precision can be achieved with short pulse laser, for surface processing in removing or ablation. Superficial texturing in form of groove-type sections with different depths and recast material, expelled on the edge of the hollow were obtained on AISI 430 stainless steel.

Laser parameters of the micro textured patterns applied have an influence of the groove morphology as concluded in [39]. The recast material is generated by the laser beam angle of attack, repetition, power, speed and frequency of the microstructuring parameters. The appearance of the recast material is especially noticeable when design type A was applied and lacks for the design type C. In case of the pattern design type B, the recast material is created but with small/insignificant values. All the patterns applied as texturing on the surface of ferritic stainless steel by a nanosecond pulsed laser point out an improvement regarding surface roughness and wettability. The presence of recast in designs A and B is linked to overlapping, due to continuous lines of the closed contours. A high overlap means that material accumulates more energy as the pulses are delivered, thus reaching and maintaining a higher temperature that leads to metal melting and rising up because of the internal pressure of the plasma plume. On the C design, having a single point, there is no overlap, so there is absolutely no energy accumulation, resulting only ablation, no recast.

The analysis of the wettability and surface roughness reveals a correlation between textured pattern design A and textured pattern design B in terms of increased contact angle and a greater surface roughness due to the recast material deposited at the edge of the crevice. Between pattern design type C and A, the link is created due to high depth of the crevice. The correlation between surface roughness and wettability can be established through frequency, speed, and number of repetitions. The higher the number of repetitions, the higher the values for the surface roughness and the contact angle at constant frequency and speed. But when the frequency and speed increase, and the number of repetitions remains constant, lower values result for surface roughness and contact angle. As consequence, the increase in roughness due to the surface texturing increases the contact angle. A similar trend was reported in [75].

Fine tuning of the superficial roughness and wettability of stainless steels finds applicability in the industry. According to the results, it is pointed out that the microtextured design type A can be easily applied in joining dissimilar materials due to an irregular surface and an increased contact area. In tribological applications, recast material can be an impediment, but in joining of dissimilar materials, this increasing of the contact area is required. The pattern design type C is suitable mainly for tribological applications, since recast material is missing.

Further investigations will be conducted to achieve super hydrophobic surfaces using the design type B optimized by diminishing the center-to-center distance of the micro texturing pattern.

Round 2

Reviewer 3 Report

The manuscript has been well improved and the only problem is the conclusion part. The conclusion should be concise and clear, in which the main work should be stated, the corresponding results and findings should be effectively summarized rather than wordy displayed.  My suggestion is that this paper could be accepted after minor revision.

Author Response

We appreciate your constructive comments and have modified the manuscript accordingly. In the second revised version the corrections made to the first revision have been highlighted in green.

  1. The manuscript has been well improved and the only problem is the conclusion part. The conclusion should be concise and clear, in which the main work should be stated, the corresponding results and findings should be effectively summarized rather than wordy displayed.  My suggestion is that this paper could be accepted after minor revision.

Whole conclusions chapter has been reformulated:

Surface texturing in form of groove-type sections with different depths and recast material, expelled on the edge of the hollow were obtained on AISI 430 stainless steel in three variants of design: 3 concentric octagonal donut patterns (type A), ellipses at an angle of 45° overlapping with its mirror (type B), respectively dimple/hole/crater array (type C).

The morphology of the textured surfaces (in terms of geometrical pattern and roughness) was linked to the laser processing parameters (spots density, processing speed) and surface wettability (contact angle).

It was observed that the roughness of the textured surface is directly proportional to the spots density (number of repetitions), while is inversely proportional to the processing speed. Surfaces with high roughness were obtained when recast material was present in significant amounts (design A) due to a higher degree of overlapping, respectively when a high crevice depth was registered in the pattern with no overlapping and recast material, as for design type C. Design B has a lower degree of overlapping, therefore an intermediate value was obtained for the surface roughness. Fine-tuning the roughness of stainless steel finds applicability in the industry. The microtextured design type A can be easily applied in joining dissimilar materials due to an irregular surface and an increased contact area. In tribological applications, recast material can be an impediment, but in the joining of dissimilar materials, this increasing of the contact area is required. The pattern design type C is suitable mainly for tribological applications since recast material is missing.

The recast material and the crevice depth of the pattern have a marked influence on the contact angle as well. The increase in roughness due to the surface texturing increases the contact angle with values ranging from 30 to 195% higher than for the neat substrate, increasing hydrophobicity. A similar trend was reported in [75].

Further investigations will be conducted to achieve super hydrophobic surfaces using the design type B optimized by diminishing the center-to-center distance of the micro texturing pattern.

Thank you,

Mircea Horia Tierean